# LanCL2 Implicates in Testicular Redox Homeostasis and Acrosomal Maturation

**DOI:** 10.3390/antiox13050534

**Published:** 2024-04-27

**Authors:** Yanling Zhao, Jichen Wang, Shuai Shi, Xinting Lan, Xiangyu Cheng, Lixia Li, Yuanfeng Zou, Lanlan Jia, Wentao Liu, Qihui Luo, Zhengli Chen, Chao Huang

**Affiliations:** 1Laboratory of Experimental Animal Disease Model, College of Veterinary Medicine, Sichuan Agricultural University, Chengdu 611130, China; zhaoyl8661@163.com (Y.Z.); saujacinthwang@163.com (J.W.); shishuaikeco@163.com (S.S.); lxt9091@163.com (X.L.); xiangyu16139@163.com (X.C.); jialanlan@sicau.edu.cn (L.J.); liuwt1986@126.com (W.L.); lqhbiology@163.com (Q.L.); 2Key Laboratory of Animal Disease and Human Health of Sichuan Province, College of Veterinary Medicine, Sichuan Agricultural University, Chengdu 611130, China; lilixia905@163.com (L.L.); yuanfengzou@sicau.edu.cn (Y.Z.)

**Keywords:** LanCL2, spermatogenesis, acrosomal maturation, redox balance

## Abstract

Redox balance plays an important role in testicular homeostasis. While lots of antioxidant molecules have been identified as widely expressed, the understanding of the critical mechanisms for redox management in male germ cells is inadequate. This study identified LanCL2 as a major male germ cell-specific antioxidant gene that is important for testicular homeostasis. Highly expressed in the brain and testis, LanCL2 expression correlates with testicular maturation and brain development. LanCL2 is enriched in spermatocytes and round spermatids of the testis. By examining LanCL2 knockout mice, we found that LanCL2 deletion did not affect postnatal brain development but injured the sperm parameters of adult mice. With histopathological analysis, we noticed that LanCL2 KO caused a pre-maturation and accelerated the self-renewal of spermatogonial stem cells in the early stage of spermatogenesis. In contrast, at the adult stage, LanCL2 KO damaged the acrosomal maturation in spermiogenesis, resulting in spermatogenic defects with a reduced number and motility of spermatozoa. Furthermore, we show that this disruption of testicular homeostasis in the LanCL2 KO testis was due to dysbalanced testicular redox homeostasis. This study demonstrates the critical role of LanCL2 in testicular homeostasis and redox balance.

## 1. Introduction

Mammalian male germ cells are produced through a continuous developmental process defined as spermatogenesis, in which undifferentiated spermatogonia undergoes mitotic expansions, meiotic divisions, and the process of spermiogenesis to generate matured spermatozoa [1,2]. Accurate control of the initiation and progress of spermatogenesis is critical for testicular homeostasis and male fertility. Spermatogenesis is unique in that it produces spermatozoa on a staggering scale, approximately 1000 sperm per second; thus, it requires a high rate of mitochondrial oxygen consumption and generates large amounts of reactive oxygen species (ROS) as byproducts of mitochondrial ATP production, correspondingly [3,4,5]. Testicular ROS level control, aiming to maintain a redox balance, appears to affect all cell types involved in spermatogenesis [6]. Dysbalanced testicular redox homeostasis could damage spermatogenic progress, with manifestations of low sperm counts, reduced sperm motility, increased sperm abnormalities, or altered fertilization, all of which result in defects in male fertility [4,6,7]. Therefore, screening and studying signals implicated in testicular redox homeostasis control could provide mechanistic insights into testicular development and male subfertility studies.

Over the decades, large numbers of cellular oxidative defense enzymes or proteins have been identified, such as superoxide dismutase (SODs), glutathione peroxidase (GPXs), peroxiredoxins (PRXs), thioredoxins (TRXs), and glutathione S-transferase (GSTs), etc. These enzymes are widely expressed in various tissues and regulated by a series of redox-sensitive transcriptional factors, playing important roles in balancing cellular redox homeostasis and protecting cells against oxidative stress [8,9]. Some polymorphisms in common antioxidant enzymes have been reported to be correlated with human male infertility [10,11,12,13,14], and the deletion of several of them resulted in knockout mice being more sensitive to external oxidative stress stimuli or age-dependent male reproductive dysfunction [15,16]. However, most male mice lacking known antioxidant genes have no defects in testicular development or fertility [16,17,18,19,20,21,22,23,24], suggesting that testicular-specific and more important oxidative defense mechanisms remain to be discovered. Previously, we described a novel redox management system mediated by the SP1-LanCL1 axis, showing the essential role of LanCL1 in maintaining neuronal and testicular redox homeostasis [25,26]. *LanCL1* has two other mammalian homologs, *LanCL2* and *LanCL3*. Among them, *LanCL3* is considered to be a pseudogene [27] and *LanCL2* correlates with doxorubicin sensitivity, abscisic acid, and the AKT/mTOR signal activity demonstrated by in vitro studies [28,29,30]. The cellular location of LanCL2 could be transferred from the plasma membrane to the nucleus by the trigger of chemical or genetic signals [27]. But, the function of *LanCL2* in vivo has not been well defined. In this study, with a genetically modified mouse model, we systematically studied the expression pattern of LanCL2 in mice, the effects of *LanCL2* deletion on testis and brain development, and the roles of LanCL2 in redox homeostasis management. We found highly enriched LanCL2 in mouse brain and testis, and development correlated with the expression of LanCL2 was noticed in the testis and brain after birth. The loss of LanCL2 did not affect normal postnatal brain development and neurobehavior but did injure testicular homeostasis induced by acrosomal maturation defects during spermiogenesis, which may have been a consequence of testicular redox imbalance.

## 2. Materials and Methods

### 2.1. Animals

All mouse work was performed according to the guidelines provided by the Animal Care and Use Committee of Sichuan Agricultural University. Conventional LanCL2 knockout mice (LanCL2 −/−, C57BL/6N background) were obtained from Cyagen Biosciences (Guangzhou, China). The mice were generated by a CRISPR-Cas9 system targeting exon 3 of LanCL2. The gRNA target sequences were as follows: gRNA1: ACT ATA TGA TAG TGA CCC TAA GG; gRNA2: ACT CTA CTA GAT CCA AAG ATG GG. Founder mice and their offspring were characterized by genotyping PCR with primers as follows: WT Forward: 5′-GCA TAG GAC AGG GTC TTA TCT AGG-3′, Reverse: 5′-CAG TGA GTC AGA TCC CCA CTT AG-3′; Cut Forward: 5′-GCA TAG GAC AGG GTC TTA TCT AGG-3′, Reverse: 5′-CAG CCC AAA TTT TCC ACC TTT CT-3′. Furthermore, the absence of the LanCL2 protein was validated by Western blots with the affinity-purified rabbit polyclonal LanCL2 antibody, which was generated by immunizing rabbits with full-length GST fusion proteins of mouse LanCL2. All mice were housed in standard, individually ventilated cages under SPF conditions with a temperature around 20–22 °C, humidity around 50–70%, and a 12 h light/12 h dark cycle. All animals were provided ad libitum access to standard chow food and water. For qRT-PCR, Western blots, and biochemical assays, the mice were euthanized through cervical dislocation and the testes and brains were dissected and frozen in liquid nitrogen immediately for experiments. For staining, the mice were perfused transcardially with phosphate buffer saline (PBS) and 4% paraformaldehyde (PFA) under narcotism first; then, the brains were stored in 4% PFA and the testes were fixed in modified Davidson’s fixative solution at 4 °C for 24 h followed by storage in 70% ethanol.

### 2.2. Quantitative Real-Time PCR

About 20 mg of each sample was used for total RNA extraction, first using the TRizol reagent (15596026, Invitrogen, Waltham, MA, USA) according to the manufacturer’s instructions. Then, ~1 µg total RNA was used for reverse transcription with the RT EasyTM II kit (with gDNase) (RT-01023, Foregene, Chengdu, China) under the following conditions: 42 °C for 25 min and 85 °C for 5 min. Then, quantitative real-time PCR was performed according to the manufacturer’s instructions with the Real Time PCR EasyTM-SYBR Green I kit (QP-01014, Foregene, Chengdu, China) for three replicates. Finally, all relative gene expressions were normalized to internal control β-Actin [31]. The primers used in this study are listed in Table 1.

### 2.3. Western Blotting

Samples were lysed by sonication with lysis buffer (2% SDS with proteinase inhibitors and phosphatase inhibitor in PBS) at 4 °C, the total protein was collected after refrigerated centrifugation, and its concentration was measured with the BCA Protein Assay Kit (23225, Thermo Scientific, Waltham, MA, USA). Then, sample lysis with 5~10 µg of protein was separated with SDS-PAGE gels, which were further electro-transferred to polyvinylidene difluoride membranes (PVDF, Millipore, Darmstadt, Germany) with the Trans-Blot^®^Turbo™ transfer system (BioRad, Hercules, CA, USA). After blocking with 5% skimmed milk in TBST (Tris-buffered saline with 0.1% Tween^®^ 20 Detergent) for 1 h, the primary antibodies were added and overnight incubation was performed at 4 °C. The membranes were washed with TBST 3 times and the secondary antibodies (1:10,000; Abclonal, Wuhan, China) were added and incubated for 1 h at room temperature. Three washes with TBST were further performed, and protein signals were finally detected and scanned using Quantitative Fluorescence Imaging Systems (ChampChemi 910, Beijing SinSage Technology, Beijing, China).

### 2.4. Immunochemical Staining

Immunohistochemical staining was performed using the SABC Kit (Boster, SA1020, Wuhan, China) according to the manufacturer’s instructions. Briefly, after being fixed with 4% paraformaldehyde solution, samples were embedded in paraffin, and microtome sections of them with a 5 μm thickness were then mounted on glass slides, which were further deparaffinized, rehydrated, and subjected to 3% H_2_O_2_ to block endogenous peroxidase activity for 20 min at RT. After 3 washes in PBS, an antigen retrieval process was performed with citrate buffer (pH 6.0) using pressure cooking, followed by a block process with blocking buffer (10% donkey serum in PBS + 0.1% Triton X-100, if a permeabilization was needed) for 1 h at RT. Then, the slides were incubated with primary antibodies (diluted in PBS with 1% donkey serum) overnight at 4 °C. After 3 washes with PBS, the biotin-labeled secondary antibody and streptavidin–biotin–peroxidase complex (SABC) were incubated on the slides in turn at RT for 30 min. Finally, color development was performed with DAB and counterstained with hematoxylin (if needed). The antibodies used in this study are listed in Table 2.

### 2.5. Immunofluorescence Staining

After the paraffin-embedded slides were deparaffinized and rehydrated, antigen retrieval was performed as described above. Slides were blocked with blocking buffer (1×PBS + 10% donkey serum + 0.01 g/mL BSA + 0.1% Triton X-100) for 60 min at RT, and primary antibodies (diluted in PBS with 1% donkey serum) were added and incubated overnight at 4 °C, followed by 3 washes with PBS for 15 min each. Then, diluted secondary antibodies were incubated in darkness for 90 min at RT. After 3 washes with PBS, the slides were mounted by coverslips with Prolong Gold with DAPI mounting medium (P36962, Invitrogen, CA, USA) and photographed under a microscope (BX61VS, Olympus, Tokyo, Japan). For the quantification of the number of different spermatogenic cells, ten microscopic fields and over 50 tubules were quantified for each testis.

### 2.6. H&E Staining and Luxol Fast Blue Staining

Hematoxylin and eosin (H&E) staining and Luxol Fast Blue (LFB) staining were performed using deparaffinized and rehydrated paraffin-embedded slides according to the manufacturer’s instructions (G1120 for H&E, G3245 for LFB, Solarbio, Beijing, China). Johnsen scoring was adopted according to the criteria listed in Table 3 [26], and, for each testis, ten microscopic fields and over 50 tubules were quantified.

### 2.7. Sperm Parameter Analysis and Acrosome Reaction Assay

Epididymis from both sides of mice were sampled and placed into Biggers–Whitten–Whittingham solution (BWW) (1 mL, 37 °C prewarmed, Table 4) [32]; the cauda epididymis was minced to make the sperms swim out. After 30 min of incubation at 37 °C, the sperm parameters were analyzed using a Fully Automatic Sperm Quality Analyzer (BX-9100A, Baoxing Medical Equipment, Xuzhou, China).

The acrosome reaction assay was performed with Coomassie bright blue staining as described before [33]. Briefly, after 30 min of incubation at 37 °C with BWW solution, the acrosome reaction of the sperm was initiated by adding calcium ionophores (Beyotime, S1672, Shanghai, China) for 1 h. The solution was centrifugated (500 g/5 min) and the precipitation was washed with PBS once. It was then centrifugated (500 g/5 min) again and resuspended with 50 μL PBS. Then, 10 μL of the solution was applied evenly on a glass slide and dried at room temperature. Then, fixation was performed with 4% PFA for 15 min, followed by staining with Coomassie solution (0.22% Coomassie Blue in 50% methanol and 10% acetic acid) for 2 min. The slides were mounted by coverslips with neutral balsam and photographed under a microscope. The crown of the sperm without acrosomal reaction was dyed dark blue; otherwise, it was colorless or light blue.

### 2.8. Cell Culture

TM3, TM4, and NCCIT cells were obtained from the National Infrastructure of Cell Line Resource (Beijing, China) and validated by short tandem repeat (STR) analysis. DMEM (PYG0073, BOSTER) containing 10% fetal bovine serum (ST30-3302P, Shanghai, PAN) was used to maintain these two cell lines in an incubator at 37 °C, with an atmosphere of 5% CO_2_. To validate its antioxidant effect, Prk5-mys-LanCL2 plasmids were transfected into NCCIT cells with TransEasyTM transfection reagent (TEO-01012, Foregene) and an indicated amount of H_2_O_2_ was used to induce oxidative stress 24 h after the transfection. Finally, Cell Counting Kit-8 was used to evaluate cell viability according to the manufacturer’s instructions (C0037, Beyotime, Shanghai, China).

### 2.9. Biochemical Assays

Samples of 20 mg were extracted in 400 μL of the recommended extraction buffer to detect the ratio of GSH/GSSG, levels of testicular MDA/PCOs and serum testosterone, and activity of SOD using biochemical assay kits according to the manufacturer’s instructions (S0053, Beyotime, for GSH/GSSG; A003-1-2, Nanjing Jianchen Bioengineering Institute, for MDA; YJ001948, mlbio, for testosterone; ml058321, mlbio, for PCOs; A001-3-2, Nanjing Jianchen Bioengineering Institute, for SOD activity). Finally, a microplate reader (AMR-100, ALLSHENG, Hangzhou, China) was used to measure the absorbance, and the values were calculated using a standard curve and used for calculating.

### 2.10. Behavioral Tests

#### 2.10.1. Mouse Gait Analysis

We used the VisuGait system (XR-FP101, Shanghai Heart Soft, Shanghai, China) for gait analysis to evaluate the motor function and coordination of the mice. Briefly, the mice were allowed to walk on a limited, straight, illuminated glass platform in a quiet environment. Below the walking platform, there were cameras capturing the footprints of the mice and pressure sensors recording the gait changes of the left front (LF), right front (RF), left rear (LH), and right rear (RH) paws. Each mouse repeated this three or more times on a straight channel. In addition, quantitative gait analysis was conducted on the main parameters.

#### 2.10.2. Accelerated Rotarod Test

An accelerated rotarod experimental device was provided by Jiangsu Cyrus Biotechnology Co., Ltd. (Xuzhou, Jiangsu, China). The mice were placed in a uniform rotating rod (rotation speed 5 r/min) with a 9 cm wide lane and a 3 cm diameter rotating rod. When the speeds of the mice were stable, they underwent a uniform acceleration process (maximum time of 5 min, speed increases every 8 s) three times. The average retention time on the revolving rod was determined. The mice were trained for three days before the experiment.

#### 2.10.3. Pole Test

Mice were placed vertically on a 50 cm tall pole with a 1 cm diameter, after which, the mice made a 180° turn and returned to the base of the pole. The mice were trained for three days before the experiment. During the test, the amount of time was recorded for the mouse to turn toward the ground (time to turn) and reach the ground (time to climb). Each mouse underwent three trials, and the average times were quantified.

### 2.11. Statistical Analysis

Data represent the mean ± standard deviation (SD) or mean ± standard error of the mean (SEM). Two-tailed Student’s *t*-test or one-way ANOVA was performed for statistical significance analysis using GraphPad Prism software (Version 9.0, San Diego, CA, USA): * *p* < 0.05, ** *p* < 0.01, *** *p* < 0.001.

## 3. Results

### 3.1. Age-Dependent Expression of LanCL2 Correlates with Testicular Maturation

A previous study showed that LanCL2 mRNA was widely expressed among tissues and enriched in the human brain and testis [34]. In this study, we also found much higher LanCL2 expression in mouse testis than in other organs at adult age, according to qRT-PCR detecting LanCL2 mRNA (Figure 1A) and Western blots analyzing its proteins (Figure 1B). We also noticed different developmental-correlated expressions of LanCL2 in mice. In the brain, LanCL2 displayed increased expression along with postnatal development (Figure 1C), while the testicular LanCL2 mRNA and protein levels maintained a robust increasing trend and paralleled the first spermatogenic wave in the first postnatal month [35,36], remaining high in adults (Figure 1D–F), which suggested critical roles of LanCL2 in regulating spermatogenic progress. To analyze the cellular localization of LanCL2 in the brain and testis, immunohistostaining was performed. We found that LanCL2 was concentrated in the body of neurons in the brain (Figure 1G) and seminiferous tubules of the testis (Figure 1H). Specifically, LanCL2 seemed to be strongly enriched in the nuclear membrane of the spermatocytes and round spermatids, and little signal was observed in Leydig or Sertoli cells (Figure 1H). Consistently, the absent LanCL2 in Leydig and Sertoli was validated by Western blot with mouse Leydig and Sertoli cell lines (TM3 and TM4 cells) (Figure 1I). The cellular localization pattern of LanCL2 proteins was consistent with the time-course results showing that the testicular expression peak of LanCL2 was around 3–4 postnatal weeks in mice (Figure 1D,F), when the spermatocytes and round spermatids make up most of the testicular cells [35]. All these results displayed a developmental-correlated expression of LanCL2 in spermatogenic maturation.

### 3.2. Loss of LanCL2 Does Not Affect Normal Brain Development

To study the biological function of LanCL2 in vivo, we generated LanCL2 knockout mice (LanCL2 −/−, KO) in which the exon 3 of LanCL2 was deleted (Figure 2A). The offspring genotypes were identified by PCR with genomic DNA (Figure 2B), and the deficiency of LanCL2 proteins was validated by Western blots (Figure 2C). Heterozygotes of LanCL2 genetically modified mice (LanCL2 +/−) were mated, and LanCL2 KO mice could be born with an expected Mendelian ratio (Figure 2D) and displayed normal postnatal viability until the age of 1.5 years. We noticed comparable body weight between LanCL2 WT and KO mice at different adult ages (Figure 2E). As LanCL2 is mostly enriched in the brain and testis of mice, we first evaluated whether the loss of LanCL2 would affect normal brain and testis development. At 12 weeks of age, LanCL2 KO mice had a normal brain index (the ratio of brain weight and body weight) compared with that of WT controls (Figure 2F). With H&E staining, normal gross morphology of the brain, lamination of the cortex, and the formation of the hippocampus could be observed in adult LanCL2 KO mice (Figure 2G–I). Also, the loss of LanCL2 seemed not to affect normal postnatal myelination in the brain, as indicated by Luxol fast blue staining (LFB staining) (Figure 2J). These data revealed that the loss of LanCL2 would not affect embryonic and following postnatal brain development.

To further verify this, we performed some behavioral tests that reflected brain function from the side. With gait analysis, we noticed normal run duration, cadence, speed, step cycle, stride length, and swing speed of LanCL2 KO mice (Figure 3A–G). Similarly, the performance of LanCL2 KO mice in the pole test and rota-rod test was comparable with that of the WT mice, indicating no difference in “time to turn” and “total time” in the pole test (Figure 3H,I) or “balance time”, “rotational speed”, and “rotations” in the rota-rod test (Figure 3J–L).

### 3.3. LanCL2 Deficiency Results in Oligoasthenozoospermia

While no observed defects were noticed in adult LanCL2 KO brains, we found a reduced testis index in LanCL2 KO mice at the age of 8 weeks (Figure 4A). Sperm parameter analysis showed an oligo-asthenozoospermic phenotype of 8-week-old LanCL2 KO mice, indicated by a robust reduction in sperm concentration, motility, and forward motility (Figure 4B–D). These defects were not a consequence of hormonal abnormalities as we detected comparable serum testosterone levels in LanCL2 KO mice (Figure 4E).

To elucidate the reason for the defects in the sperm parameters of LanCL2 KO mice, histological analysis was then performed. With H&E staining, we noticed a relatively reduced quantity and loose arrangement of spermatogenic lineage cells in seminiferous tubules of LanCL2 KO mice, which resulted in poorer Johnsen’s scores (Figure 4F,G) [37]. Then, immunohistostaining with specific markers of different testicular cell types was performed. We found that the loss of LanCL2 did not affect the number of Sertoli and Leydig cells, spermatogonia, and spermatocytes, as labeled by GATA-4, EpCAM, and SCP3, respectively (Figure 4H–K). In contrast, ACRV-1 staining that labeled acrosomes of spermatids displayed reduced positive numbers (Figure 4L). With ACRV-1 staining, we also made an assessment of spermatogenesis through the staging of seminiferous tubules (Figure 4M) according to previous methods [35,38]. We found increased seminiferous tubules from stage I to VI, in which round spermatids were present, while decreased numbers of seminiferous tubules were observed from stage VII to stage XII, where round spermatids were absent (Figure 4N). Collectively, these data suggested that LanCL2 is important for spermatogenic maturation, the loss of which could lead to spermatogenic defects and induced oligoasthenozoospermia.

### 3.4. The Loss of LanCL2 Results in the Pre-Maturation of Spermatogonia

To understand what contributed to the spermatogenic defects in the LanCL2 KO mice, we evaluated the developmental progress along with the spermatogenic process in the LanCL2 KO testis. Mammalian spermatogenesis progresses with repeating waves, and the first spermatogenic wave begins around two postnatal weeks in mice [26]. At this stage, we interestingly observed enlarged seminiferous tubules with more spermatocytes in LanCL2 KO testis, as indicated by H&E staining (Figure 5A–C) and SCP-3 immunofluorescence labeling (Figure 5D,E), which seemed to be contradictory with the decreased amount of spermatogenic lineage in adult LanCL2 KO mice. Increased spermatocytes suggested that the loss of LanCL2 could affect the rhythmicity in the self-renewal of spermatogonia. To verify this, immunolabeling with markers of spermatogonia (EpCAM) was performed, showing increased numbers of spermatogonia in 2-week-old LanCL2 KO mice (Figure 5F,G). Consistent with this, we found more phosphorylation-histone H3-positive cells in seminiferous tubules of LanCL2 KO testis at this stage, indicating enhanced cellular proliferation (Figure 5H,I). This phenotype should not include Sertoli and Leydig cells as we found no difference in the numbers of them in the LanCL2 KO testis when compared with those in WT controls at the age of two weeks (Figure 5J,K). These results revealed that the loss of LanCL2 could lead to the pre-spermatogenic maturation of mouse testis.

### 3.5. LanCL2 Is Important for Acrosomal Maturation in Spermiogenesis

Given that pre-spermatogenic maturation at the early age did not result in the fertile production of spermatozoa at the adult age, we wondered if the loss of LanCL2 affected the late process of spermatogenesis. After mitotic expansions and meiotic divisions, round spermatids are produced and undergo a sequence of morphological transformation steps like nuclear condensation, acrosome formation, cytoplasm removal, and tail elongation, all of which are defined as spermiogenesis [39]. At 4 weeks of age, during the first spermiogenic maturation process, we found remarkably fewer elongating spermatids in LanCL2 KO testis (Figure 6A,B), and this was confirmed by ACRV-1 labeling showing decreased male germ cells with acrosomes (Figure 6C,D). Developing spermatids could be divided into sixteen steps (steps 1–16) according to different acrosome structures and nuclear morphologies (Figure 6E) [40]. We noticed significantly increased numbers of spermatids in steps 1 to 4, when the acrosome was not visualized or was very small according to ACRV-1 labeling (Figure 6F). However, a reduced number of maturing spermatids was observed from steps 5 to 16, with clearly visible acrosomes (Figure 6F), suggesting that LanCL2 is important for the maturation of acrosomes. In support of this, we noticed a reduced expression of FAM71F1 and FAM209, critical factors correlated with acrosome maturation in LanCL2 KO testis (Figure 6G), as well as reduced acrosomal reactivity in LanCL2 KO spermatozoa (Figure 6H). Also, this finding was consistent with the results for the adult LanCL2 KO testis, which showed increased seminiferous tubules with round spermatids but decreased ones without round spermatids (Figure 4N). Interestingly, we found comparable numbers of spermatogonia (EpCAM-labeled) and spermatocytes (SCP-3-labeled) between LanCL2 KO mice and WT controls (Figure 6I,J,L), which were different from the phenotype observed at 2 weeks of age and suggested that the loss of LanCL2 led to pre-maturation but not increased differentiation of spermatogenic lineage cells. Also, no difference in the number of Sertoli or Leydig cells was observed, according to GATA-4 labeling (Figure 6K,L). Collectively, all these data showed that LanCL2 is critical for the progress of spermiogenesis, especially the formation and maturation of acrosomes.

### 3.6. LanCL2 Is Implicated in Regulating Spermatogenic Redox Homeostasis

As LanCL2 is a homologous gene of LanCL1 that functions as an antioxidant gene, we asked whether LanCL2 could also participate in redox homeostasis regulation. We first overexpressed LanCL2 in cultured, germ-cell-like NCCIT cells and found that it could protect against H_2_O_2_-suppressed cell viability (Figure 7A), which suggested an antioxidant effect of LanCL2. In vivo, we noticed increased GSH and decreased GSSG levels in 2-week-old LanCL2 KO testis when compared with those of WT controls (Figure 7B), indicating unbalanced redox homeostasis. In support of this, we found that the expressions of common antioxidant genes, like Prdx2 and Prdx4, increased in the LanCL2 KO testis at this stage (Figure 7C), also suggesting redox imbalance. Given that redox balance is important for the proliferation and differentiation of spermatogonial stem cells [41,42,43], redox imbalance could cause the pre-maturation of spermatogonia at the juvenal age. However, overwhelming redox imbalance always results in oxidative stress/damage and affects cellular viability. Consistently, along with testicular development, we found persistent testicular redox imbalance, indicated by reduced GSH and increased GSSG levels, in adult LanCL2 KO testis (Figure 7D), as well as suppressed expressions of common antioxidant genes (Figure 7E) and decreased testicular SOD activity (Figure 7F). In addition, we detected increased testicular malondialdehyde (MDA) and protein carbonyl (PCO) levels (Figure 7G,H), indicating oxidative damage to lipids and proteins, respectively. All these findings showed a disruption of testicular redox balance after LanCL2 deletion and implicated LanCL2 in regulating testicular redox homeostasis to participate in spermatogenic regulation.

## 4. Discussion

The regulation of spermatogenesis is complex, involving macro-level hormones and paracrine or autocrine signals generated by testicular somatic cells or germ cells from the micro-level [44,45,46]. As important signaling molecules, reactive oxygen species (ROS) affecting testicular redox homeostasis are widely involved in the regulation of processes from spermatogenesis initiation to sperm maturation in a two-edged-sword way. On the one hand, physiological levels of ROS are required for some crucial spermatogenic processes, such as spermatogonial stem cell (SSC) self-renewal. Important growth factors, like the glial-derived neurotrophic factor (GDNF) and fibroblast growth factor 2 (FGF2), could promote ROS generation to stimulate SSC self-renewal, while ROS depletion by NOX1 knockout suppresses SSC proliferation and decreases SSC numbers in mouse testis [41,42,43]. On the other hand, excessive ROS are harmful to male germ cells, containing an abundance of highly unsaturated fatty acids that are vulnerable to ROS and injure male fertility [6,47]. Therefore, knowledge describing how testicular redox homeostasis is achieved would provide benefits to understanding how spermatogenesis is accurately controlled. Previously, we reported the male germ cell-expressed antioxidant gene *LanCL1* is critical for testicular redox homeostasis [26]. In this study, we found that *LanCL2*, a homologous gene of *LanCL1*, also functioned as an antioxidant gene. Similar to LanCL1, testicular LanCL2 is a male germ cell expressed and correlated with spermatogenic maturation. At the early stage of spermatogenesis, both LanCL1 and LanCL2 exhibit low levels of expression [26,48,49]. However, the loss of either of them damages testicular redox homeostasis and causes a pre-differentiation of spermatogonia at the early stage, revealing their critical importance in maintaining male germ cells’ redox balance [26]. While the differences between these two homologs in regulating redox homeostasis in early spermatogenesis are not clear, they may complement each other, as we found increased expression of testicular LanCL1 or LanCL2 after the other one was deleted.

Along with spermatogenic progress, male germ cells lose most of their cytoplasm and, hence, have a reduced endogenous antioxidant defense [50]. Interestingly, increased expression of LanCL1 and LanCL2 was found during spermatogenic maturation, suggesting that essential roles in antioxidant defense are mediated by them during this process. However, different expression patterns of LanCL1 and LanCL2 were present in developmental testis. Both expressed in all male germ cells, LanCL1 is highly concentrated in maturing and matured spermatids [26,49], while LanCL2 is enriched in spermatocytes and round spermatids. Differences in expressions of them in different spermatogenic germ cells suggest their distinct roles during spermatogenesis. In support of this, a deficiency of LanCL1 resulted in oxidative damage and the death of maturing and matured spermatids, where it is highly expressed, thus impairing male fertility [26,49], while the loss of LanCL2 mainly caused defects of spermiogenesis, especially in the maturation of acrosomes, which are highly expressed in round spermatids. Acrosomal defects in LanCL2 KO mice were accompanied by oxidative damage and, finally, caused damage to the concentration and motility of spermatozoa. As an exocytotic organelle derived from the Golgi apparatus, acrosomes are located at the tip of the spermatozoa head, the accurate formation of which is essential for sperm maturation and further fertilization [51]. Lots of studies have elucidated the roles of physiological levels of ROS in promoting capacitation and acrosome reaction, with over-accumulated ROS causing the peroxidation of lipids and proteins, which injures acrosomal integrity and function [52,53]. However, the effects of redox balance on regulating acrosome maturation are not well defined. Sasagawa et al. showed the acrosomal expression of the antioxidant enzyme Prx4, and thought that Prx4 was involved in acrosome formation [54], while Iuchi et al. found that Prx4 knockout resulted in oxidative stress-induced spermatogenic cell death without visible acrosome defects [55]. Our results provide evidence showing that balanced spermatogenic redox homeostasis is critical for acrosomal maturation and the enriched expression of LanCL2.

While male germ cells are vulnerable to ROS-induced oxidative stress, deficiency of some common antioxidant enzymes seems not to affect their maturation and survival. The loss of SOD1 injures the fertility of female mice but not male ones, except under heat stress [15,17]; mice deficient in GSTs, PRDX1, and PRDX5 exhibit normal testicular development and male fertility [19,20,21,22,23,24]. Although the knockout of two germ-line TRXs (Txndc2 and Txndc3) impairs sperm quality, this is more likely to be a cumulative effect of age-dependent oxidative stress as defects in sperm motility and DNA integrity as well as spermatozoal oxidative damage are found only in aged mice [16]. Differently, the spermatocyte-specific deletion of GPX4 results in oligoasthenozoospermia and male infertility [56], but this is mainly because GPX4 functions as a structural protein of the mitochondrial capsule rather than an antioxidant enzyme in spermatozoa [57,58]. In contrast, deficiency of PRDX6 results in spermatozoal oxidative stress/damage and injures male fertility, while decreased spermatozoal PRDX6 levels were also found in infertile patients [10,59,60]. Our previous [26] and this current work show the correlations and differences of the family proteins LanCL1 and LanCL2 in spermatogenic redox management, which are respectively involved in the processes of maturing spermatozoal survival and acrosome maturation. Also, compared to other common antioxidant genes, these findings highlight the importance of LanCL1 and LanCL2 for spermatogenic testicular redox control and spermatozoal functions.

## 5. Conclusions

In summary, this present study provides an in-depth understanding of the correlations among LanCL2, testicular redox homeostasis, and spermiogenesis, validating the prominent role of LanCL2 in maintaining round spermatids’ redox balance, which is important for acrosomal development and sperm quality.

## Figures and Tables

**Figure 1 antioxidants-13-00534-f001:**
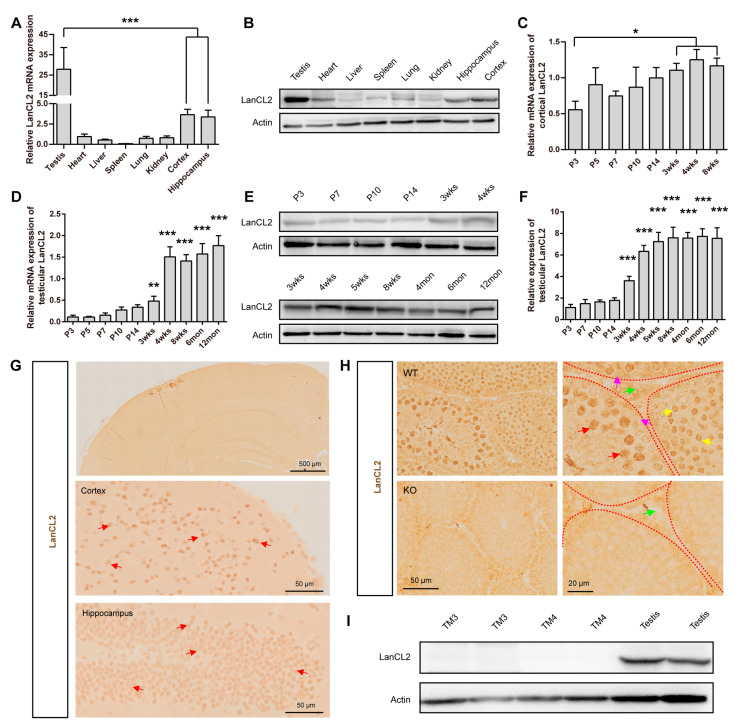
Age-dependent expression of LanCL2 correlates with testicular maturation. (**A**,**B**) Tissue expression profile of mouse LanCL2 analyzed by qRT-PCR and Western blots in 8-week-old WT mice. Error bars indicate SEM. n = 6. (**C**) qRT-PCR showing the expression profile of LanCL2 mRNA during brain development. Error bars indicate SEM. n = 3. (**D**) Immunohistostaining showing that LanCL2 is strongly concentrated in neurons of the brain. Arrows indicate positive signals. (**E**) qRT-PCR showing the expression profile of LanCL2 mRNA during testicular development. Error bars indicate SEM. n = 5. (**F**,**G**) Western blots and quantification showing the expression profile of LanCL2 protein during testicular development. Error bars indicate SD. n = 4. (**H**) Immunohistochemical images displaying enriched LanCL2 in spermatocytes and round spermatids; little signal was observed in Leydig or Sertoli cells. Red arrows indicate spermatocytes, yellow arrows indicate round spermatids, green arrows indicate Leydig cells, and purple arrows indicate Sertoli cells. (**I**) Western blots showing absence of LanCL2 in TM3 and TM4 cells. * *p* < 0.05, ** *p* < 0.01, *** *p* < 0.001.

**Figure 2 antioxidants-13-00534-f002:**
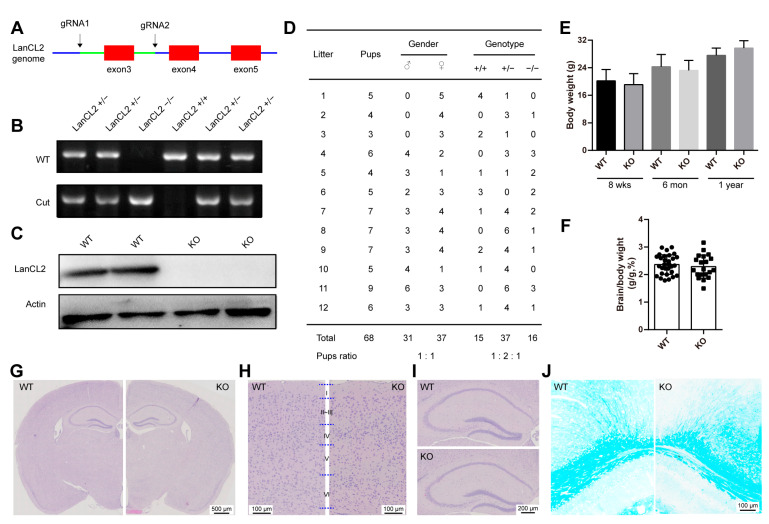
Loss of LanCL2 does not affect normal brain development. (**A**) Overview of the targeting strategy for generating LanCL2 knockout mice. Green line indicates the deleted region. (**B**) Genotyping PCR for the validation of LanCL2 knockout mice. (**C**) The absence of LanCL2 protein was validated by Western blots. (**D**) Quantification showing the number and frequency of offspring of each genotype produced by LanCL2 +/− mice. (**E**) Body weight between LanCL2 WT and KO mice at the age of 8 weeks (WT, n = 59; KO, n = 29), 6 months (WT, n = 16; KO, n = 16), and 1 year (WT, n = 5; KO, n = 7). Error bars indicate SD. (**F**) LanCL2 KO mice had normal brain index compared with WT controls. n = 33 for WT and n = 21 for KO. Error bars indicate SD. (**G**–**I**) Representative images of H&E staining showing normal gross morphology of the brain (**G**) with normal lamination of the cortex (**H**) and formation of the hippocampus (**I**) in 8-week-old LanCL2 KO mice. (**J**) Representative images of Luxol fast blue staining showing normal postnatal myelination in the brains of 8-week-old LanCL2 KO mice.

**Figure 3 antioxidants-13-00534-f003:**
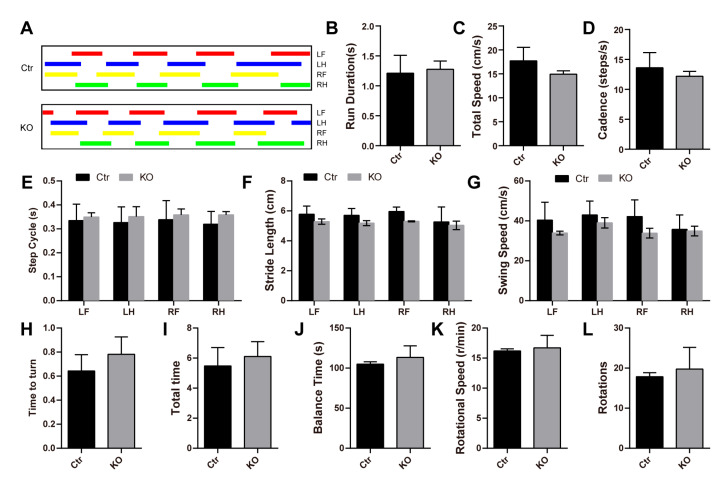
LanCL2 knockout mice do not exhibit defects in common behavioral tests. (**A**–**G**) Representative image and quantifications from gait analysis showing normal run duration, cadence, speed, step cycle, stride length, and swing speed of 8-week-old LanCL2 KO mice. Error bars indicate SD. n = 4. LF, left forepaw; LH, left hindpaw; RF, right forepaw; RH, right hindpaw. (**H**,**I**) Quantifications showing comparable performance of 8-week-old LanCL2 KO mice in pole test. Error bars indicate SD. n = 4 for Ctr, n = 5 for KO. (**J**–**L**) Quantifications showing comparable performance of 8-week-old LanCL2 KO mice in rota-rod test. Error bars indicate SD. n = 4 for Ctr, n = 7 for KO.

**Figure 4 antioxidants-13-00534-f004:**
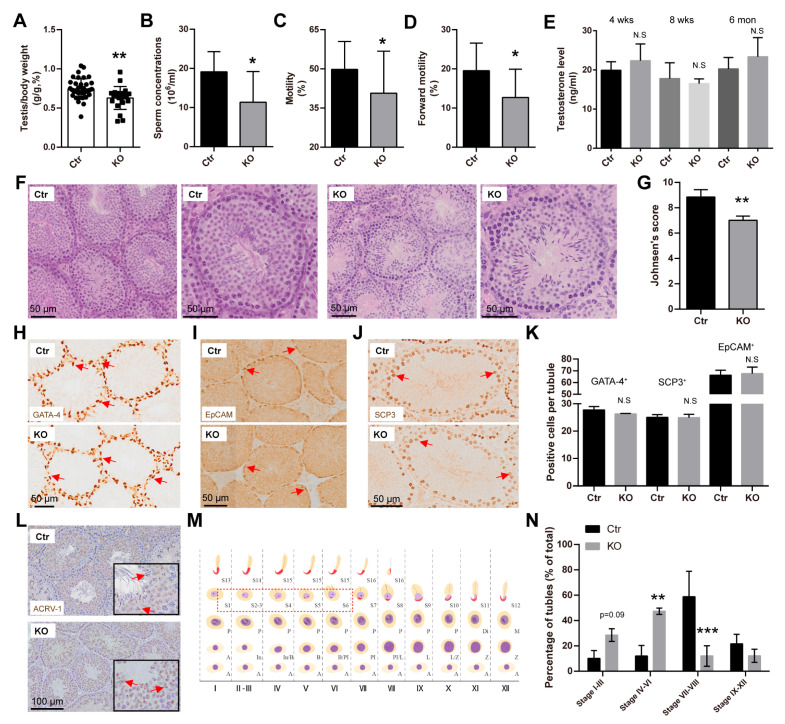
LanCL2 deficiency results in oligoasthenozoospermia. (**A**) Reduced testis index of LanCL2 KO mice at the age of 8 weeks (Ctr, n = 37; KO, n = 23). Error bars indicate SD. (**B**–**D**) Measurements showing reduced sperm concentration, motility, and forward motility (Ctr, n = 29; KO, n = 19) in 8-week-old LanCL2 KO mice. Error bars indicate SD. (**E**) Quantification showing serum testosterone levels in 4-week-old (n = 3), 8-week-old (n = 4), and 6-month-old (n = 3) LanCL2 Ctr and KO mice. Error bars indicate SEM. (**F**,**G**) Representative images of H&E staining showing reduced quantity and loose arrangement of spermatogenic lineage cells in seminiferous tubules of LanCL2 KO mice (**F**), which resulted in poorer Johnsen scores (**G**). n = 4. Error bars indicate SD. (**H**–**K**) Representative images and quantification showing labeling of GATA-4, EpCAM, and SCP-3 in 8-week-old LanCL2 Ctr and KO testis. n = 4. Error bars indicate SD. Arrows indicate positive signals. (**L**) ACRV1 immunohistochemical staining showing the acrosomes of spermatids. (**M**) Schematic representation of the 12 developmental stages of mouse seminiferous epithelium. Red dashed box shows the round spermatids. A, type A spermatogonia; In, intermediate spermatogonia; B, type B spermatogonia; Pl, preleptotene spermatocytes; L, leptotene spermatocytes; Z, zygotene spermatocytes; Di, diplotene spermatocytes; SC2, secondary spermatocyte; S1-S16, steps in spermatid differentiation. (**N**) Quantitative analysis showing increased number of seminiferous tubules from stage I to stage VI and decreased number of seminiferous tubules from stage VII to stage XII in 8-week-old LanCL2 KO mice. Error bars indicate SD. * *p* < 0.05, ** *p* < 0.01, *** *p* < 0.001.

**Figure 5 antioxidants-13-00534-f005:**
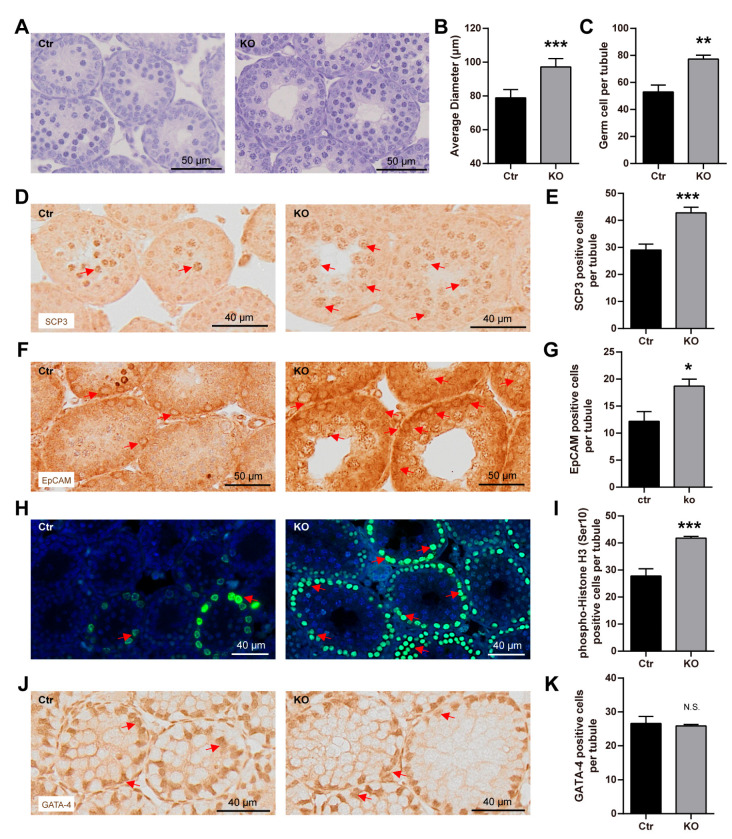
Loss of LanCL2 results in pre-maturation of spermatogonia. (**A**–**C**) Representative images of H&E staining and quantification showing enlarged seminiferous tubules with more germ cells in LanCL2 KO testis at the age of 2 weeks, n = 4. Error bars indicate SD. (**D**,**E**) Representative images of SCP-3 labeling and quantification showing increased spermatocytes in 2-week-old LanCL2 KO mice, n = 4. Error bars indicate SD. Arrows indicate positive signals. (**F**,**G**) Immunolabeling with EpCAM showing increased number of spermatogonia in 2-week-old LanCL2 KO testis, n = 4. Error bars indicate SD. Arrows indicate positive signals. (**H**,**I**) Immunofluorescence staining and quantification showing increased phosphorylation-histone H3 (Ser10)-positive cells in seminiferous tubules of 2-week-old LanCL2 KO testes, n = 4. Error bars indicate SD. Arrows indicate positive signals. (**J**,**K**) Representative images of GATA-4 labeling and quantification showing comparable Sertoli and Leydig cell numbers in 2-week-old LanCL2 KO testis. Error bars indicate SD. n = 3. Arrows indicate positive signals. * *p* < 0.05, ** *p* < 0.01, *** *p* < 0.001.

**Figure 6 antioxidants-13-00534-f006:**
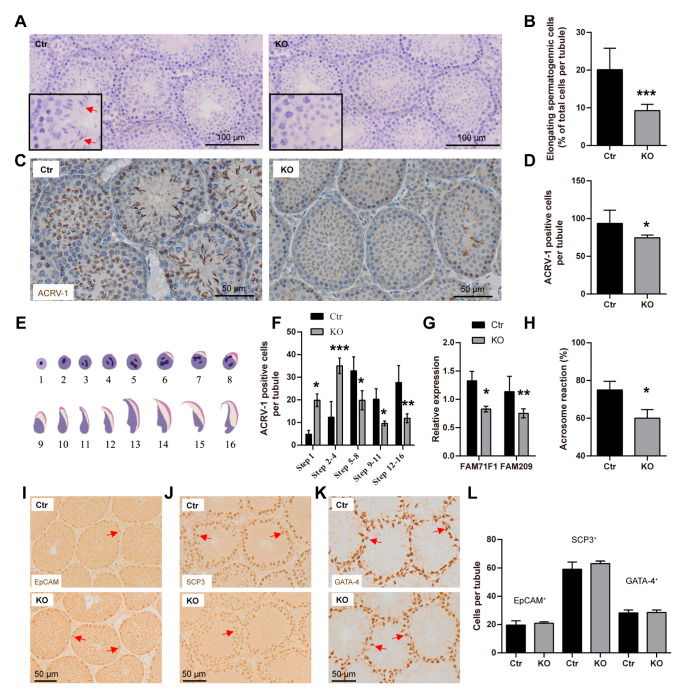
LanCL2 is important for the progress of spermiogenesis. (**A**,**B**) Representative images of H&E staining and quantification showing decreased numbers of elongating germ cells in the LanCL2 KO testis at the age of 4 weeks, n = 6 for Ctr and n = 8 for KO. Error bars indicate SD. Arrows indicate elongated spermatids. (**C**) Representative images of ACRV-1 labeling of 4-week-old LanCL2 KO testis. (**D**) Quantification showing decreased ACRV-1-positive spermatids in 4-week-old LanCL2 KO testis, n = 6 for Ctr and n = 5 for KO. Error bars indicate SD. (**E**) Illustration of the 16 developmental stages of mouse spermatids. (**F**) Quantification showing increased number of spermatids from steps 1 to 4, when the acrosome was not visualized or was very small, and decreased number of spermatids from steps 5 to 16 in 4-week-old LanCL2 KO testis, n = 3. Error bars indicate SD. (**G**) qRT-PCR showing reduced expressions of FAM71F1 and FAM209 in 4-week-old LanCL2 KO testis. Error bars indicate SEM. n = 3 for Ctr and n = 4 for KO. (**H**) Quantification showing reduced acrosomal reactivity of LanCL2 KO spermatozoa. n = 3. (**I**–**L**) Representative images of EpCAM (I), SCP3 (J), and GATA-4 (K) labeling and quantification showing comparable numbers of spermatogonia, spermatocytes, and Sertoli and Leydig cells between LanCL2 Ctr and KO mice at the age of 4 weeks. Arrows indicate positive signals. Error bars indicate SD. * *p* < 0.05, ** *p* < 0.01, *** *p* < 0.001.

**Figure 7 antioxidants-13-00534-f007:**
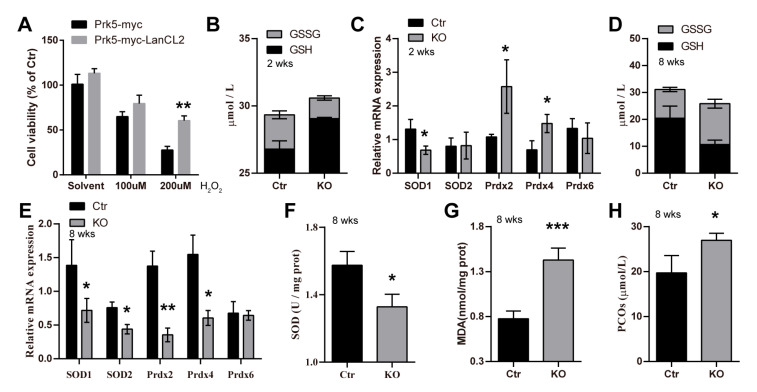
LanCL2 is important for spermatogenic redox homeostasis. (**A**) Overexpression of LanCL2 in NCCIT cells can significantly improve cell viability treated with H_2_O_2_. (**B**) Levels of testicular GSSG (*p* = 0.0062) and GSH (*p* = 0.0032) at the age of 2 weeks, n = 3. (**C**) qRT-PCR showing the expressions of common antioxidant genes in 2-week-old LanCL2 Ctr/KO testes, n = 3. (**D**) Levels of testicular GSSG (*p* = 0.0126) and GSH (*p* = 0.0237) in 8-week-old mice, n = 3. (**E**) qRT-PCR showing the expressions of common antioxidant genes in 8-weeks-old LanCL2 Ctr/KO testes, n = 3. (**F**) Quantification showing the testicular SOD activity at the age of 8 weeks, n = 10 for Ctr and n = 9 for KO. (**G**) Quantification showing testicular MDA levels at the age of 8 weeks, n = 11 for Ctr and n = 9 for KO. (**H**) Quantification showing testicular PCOs at the age of 8 weeks, n = 4 for Ctr and n = 3 for KO. Error bar indicates SEM. * *p* < 0.05, ** *p* < 0.01, *** *p* < 0.001.

**Table 1 antioxidants-13-00534-t001:** Primers for qRT–PCR.

Gene	Primer	Sequence
β-actin	F	5′-AGA GGG AAA TCG TGC GTG AC-3′
	R	5′-CAA TAG TGA TGA CCT GGC CGT-3′
LanCL2	F	5′-GTG TAG CGA TGT GAT TTG GC-3′
	R	5′-AAT GCT GGA AAC CGT GAT GT-3′
Prdx2	F	5′-CAT TCC AGT TCT CGC TGA CA-3′
	R	5′-GTT TTG TGA TGG GTC GAT GA-3′
Prdx4	F	5′-AGA GGA GTG CCA CTT CTA CG-3′
	R	5′-GGA AAT CTT CGC TTT GCT TAG GT-3′
Prdx6	F	5′-GAC TCA TGG GGC ATT CTC TTC-3′
	R	5′-CAA GCT CCC GAT TCC TAT CAT C-3′
SOD1	F	5′-AAC CAG TTG TGT TGT CAG GAC-3′
	R	5′-CCA CCA TGT TTC TTA GAG TGA GG-3′
SOD2	F	5′-TGG ACA AAC CTG AGC CCT AAG-3′
	R	5′-CCC AAA GTC ACG CTT GAT AGC-3′
FAM71F1	F	5′-ATG ATG ACA TCA GTT CCA CCT AGA AAG TC-3′
	R	5′-TAT AGA GTT TCC TCC AGT TAG GGA CAG CC-3′
FAM209	F	5′-TGC CTG TTC TTG TCT CTG TG-3′
	R	5′-TCA CCA ATT CCA TCT CGA GC-3′

**Table 2 antioxidants-13-00534-t002:** Antibodies used in this study.

Antibodies	Source	Identifier/Application
EpCAM Rabbit mAb	ABclonal (Wuhan, China)	Cat#A19301, IHC: 1:500
GATA-4(D3A3M) Rabbit mAb	Cell Signaling Technology (Danvers, MA, USA)	Cat#36966, IHC: 1:800
ACRV1 Rabbit pAb	Proteintech (Wuhan, China)	Cat#14040-1-AP, IHC: 1:500
SCP3 Rabbit pAb	Abcam (Cambridge, MA, USA)	Cat#ab15093, IHC: 1:500
Phospho-Histone H3-S10	ABclonal (Wuhan, China)	Cat#AP0002, IF: 1:200
LanCL2 Rabbit pAb	Generated by immunizing rabbits with full-length GST fusion proteins of mouse LanCL2	IHC: 1:250, WB: 1:50
β-Actin Rabbit mAb	Abclonal (Wuhan, China)	Cat#AC026, WB: 1:100,000
aRab-488 Alexa Fluor	Jackson ImmuneResearch (West Grove, PA, USA)	711-547-003, IF: 1:50
aRab-594 Alexa Fluor	Invitrogen (Carlsbad, CA,USA)	A-11037, IF: 1:500
aGoat-488 Alexa Fluor	Invitrogen (Carlsbad, CA,USA)	A-11055, IF: 1:1000
aM-594 Alexa Fluor	Invitrogen (Carlsbad, CA,USA)	A-21203, IF: 1:1000

**Table 3 antioxidants-13-00534-t003:** Johnsen score criteria.

Score	Morphology
10	Complete spermatogenesis
9	Slightly damaged spermatogenesis, large number of late-stage sperm cells, and disorder of spermatogenic epithelium
8	Less than 5 sperm in the lumen and a small amount of late-stage sperm cells
7	Mo sperm or late-stage sperm cells in the lumen but a large number of early-stage sperm cells
6	No sperm or late-stage sperm cells in the lumen with a small amount of early-stage sperm cells
5	No sperm or sperm cells in the lumen but a large number of spermatocytes
4	No sperm or sperm cells in the lumen with a small amount of spermatocytes
3	Only spermatogonia in the lumen
2	Only Sertoli cells in the lumen and no germ cells
1	No seminiferous epithelium

**Table 4 antioxidants-13-00534-t004:** Biggers–Whitten–Whittingham solution formula.

Component	Concentration
NaCl	95 mM
NaHCO_3_	25 mM
HEPES	10 mM
Glucose	5 mM
KCl	4.8 mM
Lactic acid	2.0 mM
CaCl_2_	1.3 mM
MgSO_4_	1.2 mM
KH_2_PO_4_	1.2 mM
Na-pyruvate	0.25 mM
BSA	3 mg/mL

## Data Availability

The original data of the current study are available from the corresponding authors upon reasonable request.

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
