# Peer review of "LanCL2 Implicates in Testicular Redox Homeostasis and Acrosomal Maturation"

_antioxidants, 2024, doi:10.3390/antiox13050534_

Round 1
Reviewer 1 Report
In this paper, the authors characterized the LanCL2 protein in male gametogenesis. They found
that LanCL2 expression correlates with testicular maturation and is particularly expressed in spermatocytes and round spermatids. In addition, LanCL2 KO caused a precocious maturation and accelerated the self-renewal of spermatogonial stem cells while, at the adult stage, LanCL2 KO damaged the acrosomal maturation. Finally, they showed that these effects may be due to imbalanced testicular redox homeostasis.
The paper is interesting and overall well-conducted: here are some comments that may improve its quality:
- In the introduction, more information about LANCL2 function should be provided, for example by specifying its cellular sub-localization on the plasma membrane and in the nucleus;
- In the Materials and Methods section, some information is completely missing and should be added, such as the number and the age of the used animals, as well as how they were sacrificed and the samples collected and stored;
- Sentences in lines 110-111 and 181-182 should be rephrased (it seems instructions that can be read in a protocol);
- Information on the used optical microscope should also be provided;
- Section 2.8: why just the NCCIT cell line was used? They should have considered for example GC-1 and/or GC-2 cells, since are spermatogenic cell lines;
- Lines 201-202: why some data sets were expressed as mean ± SD and others as mean ± SEM?
- Line 209: the authors did not confirm the cited work, since Mayer et al. used human samples;
- Line 219: the authors should specify the nuclear localization of LanCL2 and consider this aspect also in the Discussion;
- Line 220: the lower part of Fig. 1H, showing KO, is not cited in the relative paragraph;
- Fig. 4L is too small and spermatids are not appreciable;
- Fig. 5H seems blurred and out of focus, their quality should be improved;
- Since the authors suggested the role of LanCL2 on acrosome formation, an in vitro acrosome reaction should be performed to strengthen this point.
- Throughout the text, the meaning of the used acronyms should be specified at their first mention.
Author Response
The response to reviewer 1 is attached.

Reviewer 2 Report
The manuscript is very interesting but there are several weaknesses.
1) The information is too numerous. The authors force graphs and photos into single figures that are hard to read (the same for the legend). They should consider dividing the manuscript into two separate papers.
2) the English form is in many cases approximate. Verbs are often in the wrong tense and several sentences have no meaning. For example, lines 49-51, or 58 or 105. A revision is necessary.
3) in the results there are often references to previous works or considerations that should be included in the discussion
4) the methods lack references for the techniques indicated. Furthermore, there is no description of how the qualitative information (for example, in Figure 4) was obtained. It is stated that the number of cells increases/decreases, but it is impossible to understand how counts were done (and analysed statistically).
5) There are some inconsistencies in the results that need to be clarified. The position of the labelling, for example, changes after the KO (Figure 5F).
6) A few minor editing mistakes. For example, the authors write ‘Figure. 4A’ instead of Figure 4A’. Also, a few spaces missing or extra: (line 164: 37°C; 171, MDA / PCO and elsewhere). In addition, I suggest dividing the text into shorter paragraphs: it is very difficult to retrieve information in a long text. For example, in line 297: the text moves from data on motility and number of sperm to histological data. If authors introduce a new paragraph, the reader immediately realises that the topic changed.
The authors must resolve these points before examining the discussion.
Below there is a list of more specific observations.
INTRODUCTION
Line 62: what is intended for ‘the effects of LanCL2? In the following lines, there is no mention of the behavioural tests. The aims, therefore, need some attention.
MATERIALS AND METHODS
All the section lacks references.
Line 76: it is uncommon to refer to results in methods. Can revise?
Line 80: It is meant ‘the absence of LanCL2?
Line 88: 20mg. introduce a space.
Line 96: eliminate as follows.
Table 1. it would be great if references were added to the primers list.
Line 102: specify the buffer.
Line 109: concentration of primary antibodies?
Line 112: TBST stands for?
Line 118: sections are mounted on slides. Please amend.
121: under high pressure? Specify.
Line 125: SABC is…
Line 125-126: sections are incubated with secondary antibody and DAB not vice versa. Correct.
Line 126: explain why using DAB
Line 136. Antibodies were fluorescent. Which fluorochrome was used?
Line 145: The Jonsen score was adopted, not performed. For criteria, mention references (or, if considered necessary, add as an extra material).
Line 155: use a reference instead of listing components.
No info on quantitative analyses in the sections. How were they performed? How many sections? Examined all sections or portions? How was tested statistical significance?
RESULTS
Lines 208-216: very confused. The authors refer to previous data and include references, mix testis and brain, and different techniques. Reorganise.
Line 212: figure 1C says that the increase is not slight but what I see is a significant increase.
Legend Figure E does not mention testis (while the legend of Figure C does).
Graphs are so close that Y-axis titles fuse. Blot is at one level, quantification at another. The section and the figure are impossible to follow. Reorganise.
Figure D: needed a low magnification picture to show the entire cortex labelled (and possibly, other negative/positive parts of the brain)
Figure F and D: why in Figure F labelling is on the cytoplasm and in the testis on the chromosomes? Where is located LanCL2 protein? If nuclear, is chromosomal?
The same is observed in Figure 5F. In control mice, labelling is on the cytoplasm; in KO mice in the nucleus. The text should highlight the different locations.
Line 259: The authors state that ‘loss of LanCL2 would not affect embryonic and the following postnatal brain development. But observing Figure 2H it appears that the WT animals have a much better-developed cortex than Ko mice since the number of neurons per unit of area is markedly reduced. A more detailed evaluation of the cortical condition is necessary.
Line 279: the authors affirm that they see a ‘normal postnatal myelination’. Though apparently identical, the KO mice have less developed myelination.
Separate different topics in different sub-paragraphs. It is difficult to retrieve different sets of information in a 23-line text.
Line 253: Brain index: how was calculated? No mention (if I checked well) in methods. Was a ratio calculated on fresh tissue?
The lack of behavioural effects seems to be ‘masked’ by the high SD of controls. In addition, have proven that all the cortex is normally developed? In a few words: all the cortex is properly developed? If not, different functions may be differently impaired.
Figure 4B-D. It is hard to believe that WT and KO data are different considering the huge SD.
Figure 4F: there is an apparent loss of the basal membrane around the tubules. Can you confirm? explain why? and, in case, explain if this might have impacted on tubule development.
The legends, in general, are very long and difficult to follow (due to the large number of figures present). It is possible to simplify them? For example, avoid repeating that SD stands for standard deviation. add the info once, at the end.
Line 300 and 306: how quantity was determined? How were normalised data from different experiments? Add info in methods.
Line 320 and 323. Why SD and SEM? Explain why reporting two different data.
Lines 336 and 339. The authors should move these sentences into discussion.
Figure 6: methods to obtain quantitative data must be explained.
Line 413: LanCL2 ‘is implicated in regulation’ is probably more correct.
INTRODUCTION
Line 62: what is intended for ‘the effects of LanCL2? In the following lines, there is no mention of the behavioural tests. The aims, therefore, need some attention.
MATERIALS AND METHODS
All the section lacks references.
Line 76: it is uncommon to refer to results in methods. Can revise?
Line 80: It is meant ‘the absence of LanCL2?
Line 88: 20mg. introduce a space.
Line 96: eliminate as follows.
Table 1. it would be great if references were added to the primers list.
Line 102: specify the buffer.
Line 109: concentration of primary antibodies?
Line 112: TBST stands for?
Line 118: sections are mounted on slides. Please amend.
121: under high pressure? Specify.
Line 125: SABC is…
Line 125-126: sections are incubated with secondary antibody and DAB not vice versa. Correct.
Line 126: explain why using DAB
Line 136. Antibodies were fluorescent. Which fluorochrome was used?
Line 145: The Jonsen score was adopted, not performed. For criteria, mention references (or, if considered necessary, add as an extra material).
Line 155: use a reference instead of listing components.
No info on quantitative analyses in the sections. How were they performed? How many sections? Examined all sections or portions? How was tested statistical significance?
RESULTS
Lines 208-216: very confused. The authors refer to previous data and include references, mix testis and brain, and different techniques. Reorganise.
Line 212: figure 1C says that the increase is not slight but what I see is a significant increase.
Legend Figure E does not mention testis (while the legend of Figure C does).
Graphs are so close that Y-axis titles fuse. Blot is at one level, quantification at another. The section and the figure are impossible to follow. Reorganise.
Figure D: needed a low magnification picture to show the entire cortex labelled (and possibly, other negative/positive parts of the brain)
Figure F and D: why in Figure F labelling is on the cytoplasm and in the testis on the chromosomes? Where is located LanCL2 protein? If nuclear, is chromosomal?
The same is observed in Figure 5F. In control mice, labelling is on the cytoplasm; in KO mice in the nucleus. The text should highlight the different locations.
Line 259: The authors state that ‘loss of LanCL2 would not affect embryonic and the following postnatal brain development. But observing Figure 2H it appears that the WT animals have a much better-developed cortex than Ko mice since the number of neurons per unit of area is markedly reduced. A more detailed evaluation of the cortical condition is necessary.
Line 279: the authors affirm that they see a ‘normal postnatal myelination’. Though apparently identical, the KO mice have less developed myelination.
Separate different topics in different sub-paragraphs. It is difficult to retrieve different sets of information in a 23-line text.
Line 253: Brain index: how was calculated? No mention (if I checked well) in methods. Was a ratio calculated on fresh tissue?
The lack of behavioural effects seems to be ‘masked’ by the high SD of controls. In addition, have proven that all the cortex is normally developed? In a few words: all the cortex is properly developed? If not, different functions may be differently impaired.
Figure 4B-D. It is hard to believe that WT and KO data are different considering the huge SD.
Figure 4F: there is an apparent loss of the basal membrane around the tubules. Can you confirm? explain why? and, in case, explain if this might have impacted on tubule development.
The legends, in general, are very long and difficult to follow (due to the large number of figures present). It is possible to simplify them? For example, avoid repeating that SD stands for standard deviation. add the info once, at the end.
Line 300 and 306: how quantity was determined? How were normalised data from different experiments? Add info in methods.
Line 320 and 323. Why SD and SEM? Explain why reporting two different data.
Lines 336 and 339. The authors should move these sentences into discussion.
Figure 6: methods to obtain quantitative data must be explained.
Line 413: LanCL2 ‘is implicated in regulation’ is probably more correct.
Author Response
The response to reviewer 2 is attached

Reviewer 3 Report
The authors investigated the function of LanCL2 in spermatogenesis and brain development. They revealed that LanCL2 have role in testicular redox homeostasis and acrosomal maturation with no effects on brain development. The current study consisted of extensive and detailed experiments, and conclusions seem appropriate. However, there are many points to be elucidated for the publication.
There are some English grammar errors: e.g.) “showing”, ”have” in Line 60, “provide” in Line 71, and others.
Line 58, 60 and others: What is the difference between “LanCL2” and “Lan (Italic) CL2”?
Line 63: Conclusive descriptions should not be placed in “Introduction”.
Line 76: Figures must be numbered in the order they appear in the text.
Line 88: It is not stated how animal samples were collected including method of euthanasia.
Line 217: “neuron” consists of cell body, dendrite and axon. What the authors indicated here would be just the cell bodies (Fig. 1D).
Line 306: How did the authors define the positive and negative cells? What was the threshold? At least, it is impossible to define them in Fig. 4L.
Line 327-328: the explanation for Fig. 4M is not enough and it is hard to understand what it shows.
Line 343: Same as mentioned in Line 306. Especially stanning background in Fig. 5F is so high that it is hard to define.
Line 359-:362: Why the fluorescence was used only in this immunostaining? Fig. 5H is not in focus.
Line 404: “Stage” is used in the graph of Fig. 6F instead of “step”.
Line 439: X-axis item is unclear (no explanation even in Materials and methods).
Line 451-468: Such background information should be described in “Introduction”, not “Discussion”.
Line 493-495: Are the LanCL2 mice infertile?
Line 523: “In summary” seems to be redundant in “Conclusion” section. The phase “this present study” is also redundant.
Author Response
The response to reviewer 3 is attached

Round 2
Reviewer 1 Report
The authors addressed all the raised issues. The paper can be accepted for publication in its form.
The authors addressed all the raised issues. The paper can be accepted for publication in its form.
Author Response
Thanks for the reviewer's efforts on evaluating our work.
Reviewer 2 Report
Accepted
No indications
Author Response

(The authors gave the same response as above.)
